# Simultaneous Enantiodivergent Synthesis of Diverse Lactones and Lactams via Sequential One-Pot Enzymatic Kinetic Resolution–Ring-Closing Metathesis Reactions

**DOI:** 10.3390/molecules27227696

**Published:** 2022-11-09

**Authors:** Anna Brodzka, Dominik Koszelewski, Ryszard Ostaszewski

**Affiliations:** Institute of Organic Chemistry PAS, Kasprzaka 44/52, 01-224 Warsaw, Poland

**Keywords:** enzymatic kinetic resolution, ring-closing metathesis, lactones, lactams, diversity-oriented synthesis

## Abstract

One of the goals of diversity-oriented synthesis is to achieve the structural diversity of obtained compounds. As most biologically active compounds are chiral, it is important to develop enantioselective methods of their synthesis. The application of kinetic resolution in DOS is problematic because of low efficiency (max. 50% yield) and many purification steps. The further derivatization of kinetic resolution products in DOS leads to the formation of a narrow library of compounds of the same stereochemistry. To overcome these limitations, we present a new concept in which the kinetic resolution is combined, the subsequent reaction of which in a one-pot protocol leads to the simultaneous formation of two skeletally and enantiomerically diverse platform molecules for DOS. Their further derivatization can gain access to a double-sized library of products in respect to a classical approach. The validity of our concept was evidenced in enzymatic kinetic resolution followed by a ring-closing metathesis cascade. From racemic carboxylic acid ester, a simultaneous formation of enantiopure lactones and lactams was achieved. These compounds are important building blocks in organic and medicinal chemistry and until now were synthesized in separate procedures.

## 1. Introduction

Over the past two decades, diversity-oriented synthesis (DOS) has emerged as one of the main tools in drug discovery and medicinal chemistry. It provides quick access to a small collection of skeletally varied organic compounds starting from simple and similar substrates [1,2,3,4,5]. In general, the DOS strategy involves the development of a chemical transformation leading to the formation of one functionalized product, which in the next steps can be subjected to various reactions yielding several skeletally distinct compounds. Thus, the products of each step become the branching substrates for subsequent steps.

As a great amount of pharmaceuticals and biologically active compounds are chiral, and thus their activity is strongly enantiomer-dependent, it is extremely important to develop the stereodiverse libraries of compounds [6]. Usually, for the stereoselective DOS, stereocontrolled reactions are applied resulting in the formation of the range of products with the same configuration of the chiral center [7,8,9].

Kinetic resolution (EKR) is a well-known ad very useful method for the synthesis of enantiomerically pure compounds [10,11,12]. However, the usability of such transformations can be limited in diversity-oriented approach, mainly because of the maximum conversion of 50%. There are few reports concerning the application of kinetic resolution reaction for DOS [13,14,15,16]. Nevertheless, in those examples only a maximum 50% of the staring material is subjected to the subsequent reactions, or the unreacted enantiomer is then converted to the same product with the opposite configuration and, after that, subjected to the subsequent reactions used in DOS (Figure 1A). However, the product of kinetic resolution and its unreacted substrate have to be chromatographically resolved prior to the next transformation. Even if the unreacted enantiomer is chemically transformed into the product of kinetic resolution (with opposite stereochemistry), further derivatization of both enantiomers in n DOS reactions will lead to the library of n structurally diverse compounds with opposite stereochemistry (Figure 1A). Thus, to overcome this limitation, we propose the new concept based on the combination of kinetic resolution of racemic substrate with the subsequent reaction in a one-pot cascade (Figure 1B). The optically pure compounds (product and unreacted substrate) obtained via kinetic resolution would be subsequently (without isolation and purification) derivatized in the reactions commonly used in DOS, leading to the simultaneous formation of two skeletally and enantiodivergent building blocks. Their further derivatization via n DOS reactions can gain access to the library of 2n compounds with diverse stereochemistry. A twofold larger library of structurally and stereochemically diverse compounds can be obtained in shorter time and steps. Such an approach oversteps the existing limitations of stereoselective DOS.

To verify the proposed concept, we combined an enzymatic variant of kinetic resolution reaction with a subsequent ring-closing metathesis as the second reaction (Figure 1). 

One of the most important classes of compounds from the medicinal chemistry and pharmaceutical point of view is the chiral small ring lactones and lactams [17,18,19]. Despite their high potential for application in medicinal chemistry, they are often excluded from drug discovery programs as being synthetically challenging [20,21,22]. Thus, synthesis and screening of small molecule libraries containing diverse enantiopure lactones and lactams will be highly valuable. Among others, chiral enantiopure small ring endocyclic enol lactones are of particular interest because their scaffolds are present in many biologically active natural products [23,24,25,26] and they are important building blocks for the synthesis of many pharmaceutically relevant compounds. However, the literature syntheses of optically pure monosubstituted six-membered endocyclic enol lactones are rather scarce [27]. Feringa et al. described the catalytic asymmetric conjugate addition of alkyl Grignard reagents to pyranones leading to the formation of lactones [27]. Despite the high enantioselectivity of presented methodology, this protocol allows to obtain only alkyl-substituted lactones.

Recently, we have developed the efficient synthesis of valuable synthons of endocyclic enol lactones via ring-closing metathesis starting from vinyl esters [28] or carboxylic acids [29]. However, these methodologies allow us to obtain only racemic products. 

We have also developed the synthetic methodology toward optically pure δ-lactones based on the enzymatic kinetic resolution (EKR) and ring-closing metathesis reactions [30]. However, the main drawback of EKR is that the maximum of theoretical yield is 50%. Thus, to overcome this inconvenience, we have also elaborated a sequential, two-step chemo-enzymatic procedure toward 5,6-dihydropyran-2-ones based on enzymatic dynamic kinetic resolution, followed by ring-closing metathesis [31]. Unfortunately, such a concept cannot be applied for the synthesis of endocyclic enol lactones. 

There are only few reports considering the synthesis of chiral enantio-enriched small or medium-sized lactones or lactams [32,33,34]. The development of novel diversity-oriented synthesis of chiral enantiopure six- or seven-membered lactones and lactams is of considerable interest. Thus, on the basis of the DOS approach and our experience in enzymatic protocols [35,36,37,38,39,40] and RCM reactions [28,29,30,31], we proposed the chemoenzymatic approach toward structurally and stereogenically divergent lactones and lactams. This approach is based on the enzymatic kinetic resolution of chiral vinyl esters via enzymatic transesterification or amidation, followed by ring-closing metathesis leading to the formation of enantiopure lactones and lactams with opposite stereochemistry (Figure 1). The presented approach would be an extension of diversity-oriented synthesis, as two skeletally and enantiomerically distinct products will be obtained in a one-pot cascade. The formation of two valuable enantiomerically pure distinct products in a one-pot protocol makes this procedure environmentally benign. As both enantiomers of the substrate are converted into products, the *E*-factor [41], defined as the amount of waste produced per amount of unit, decreases dramatically. Moreover, the proposed transformation is atom economic.

The obtained six- or seven-membered lactones and lactams are important compounds from the medicinal chemistry point of view. For example, enantiomerically pure 4-phenyl-3,4-dihydro-2H-pyran-2-one can serve as a platform molecule for the synthesis of a library of multiple skeletally diverse biologically active compounds. These include phenibut, a neurophychotropic drug, a GABA antagonist used in the treatment of anxiety, insomnia, depression, alcoholism, and many other ailments [42,43]. ε-Lactam motifs are widely found in many active pharmaceutical ingredients (API) and biologically relevant compounds [44].

Herein, we report the results of our studies on the one-pot stereoselective synthesis of valuable lactones and lactams, which can serve as the platform molecules for the divergent synthesis of various structurally diverse compounds.

## 2. Results and Discussion

To verify the proposed concept, as an example of a kinetic resolution-subsequent reaction cascade, we studied the combination of enzymatic kinetic resolution with ring-closing metathesis. We began our studies from the development of the enzymatic kinetic resolution step. The main limitation of this approach is enantioselectivity. Thus, the careful choice of reaction conditions is essential. Vinyl esters are frequently used as selective acyl donors in enzymatic kinetic resolution (EKR) of various chiral alcohols and amines because the formed acetaldehyde shifts the reaction equilibrium in favor of the products [45,46]. Thus, we have tested if enzymatic transesterification of vinyl ester of chiral carboxylic acid with achiral alcohol would lead to efficient kinetic resolution. As a model substrate, we chose racemic vinyl 3-phenyl-4-pentenoate (**1a**), which was transformed into allyl ester (**3a**) in the reaction with allyl alcohol (**2**) catalyzed by various biocatalysts. As the vinyl ester **1a** and allyl ester **3a** are inseparable, either by flash chromatography or by HPLC, the obtained esters were subsequently transformed into easily separated lactones **4a** and **5a** via our previously developed methodology [28]. The initial experiments were carried out in toluene at 60 °C as this solvent was the most suitable for the ring-closing metathesis reaction. The first step, enzymatic transesterification, was performed over 24 h; after that, the enzyme was filtered off and the solvent was evaporated. The residue was dissolved in another portion of toluene and Grubbs second-generation catalyst was added (Table 1). The results are summarized in Table 1. We have tested a vast number of commercially available biocatalysts (native and immobilized) and several domestic-made liver acetone powders. However, only several of the tested enzymes catalyzed the studied transformation (Table 1), and only those results are presented.

Among the tested biocatalysts, the catalytic effect was observed only for six enzymes: Amano AK lipase from *Pseudomonas fluorescence* (Amano AK PfL), Amano PS lipase from *Pseudomonas cepacia* (Amano PS PcL), lipase from *Rhizopus oryzae* (RoL), lipase B from *Candida antarctica* (CALB), Novozym, and Lipozyme (Table 1). The transesterification reaction in the absence of enzyme does not proceed, and, after the metathesis, only the racemic lactone **4a** was formed (Table 1, entry 1). The lowest efficiency of KR reaction was observed for lipase from *Rhizopus oryzae* as product **5a** was obtained in 6% yield as a racemate (Table 1, entry 3). When the reactions were catalyzed by lipases Amano PS PcL and CALB, the desired lactone **5a** was formed with slightly better yields and with moderate to good enantiomeric excesses (Table 1, entries 3 and 5). The application of immobilized enzymes, Amano AK PfL and Lipozyme, resulted in the formation of product **5a** with the yields near 40%; however, the enantioselectivities of studied transformations were low (Table 1, entries 2 and 7). The best results were obtained for Novozym, the six-membered enol lactone **4a** was isolated in 65% of yield and 42% *ee* and the corresponding seven-membered lactone **5a** was isolated as a single enantiomer (*R*) in 29% yield (Table 1, entry 6). Thus, Novozym was selected for the further studies.

As a next set of experiments, we studied the influence of solvents and temperature on the enzymatic kinetic resolution of vinyl 3-phenyl-4-pentenoate (**1a**) with allyl alcohol (**2**) catalyzed by Novozym. As previously, the results were measured after the transformation of obtained esters to corresponding lactones **4a** and **5a**. The results are summarized in Table 2.

The enzymatic reactions conducted in toluene at all tested temperatures (from 40 to 70 °C) provided the seven-membered lactone (*R*)-**5a** in an enantiopure form (Table 2, entries 1–4). We have observed that the efficiency of transesterification reaction was related with the temperature of reaction. The best result was obtained at 70 °C (Table 1, entry 4). The change of solvent on volatile *tert*-butyl methyl ether (TBME) resulted in an increase in yield and a decrease in an enantioselectivity (Table 2, entry 5). When the enzymatic reactions were conducted in dichloromethane (DCM), tetrahydrofuran (THF) or chloroform, the formation of products was not observed (Table 2, entries 6–8). When acetonitrile was used as a solvent at 40 °C, the product **5a** was isolated in moderate yield and low enantioselectivity (Table 2, entry 9). Thus, the optimal reaction conditions are toluene at 70 °C. When the EKR reaction time was extended to 48 h, both lactones were obtained as single enantiomers: (*S*)-**4a** and (*R*)-**5a** (Table 2, entry 10). We have also tested if analogues of allyl alcohol can be used as alkoxy group donors for the enzymatic transesterification of **1a**. However, the results were worse than for allyl alcohol. Thus, the optimal reaction conditions for the enzymatic kinetic resolution of vinyl 3-phenyl-4-pentenoate are Novozym, toluene at 70 °C during 48 h and allyl alcohol as an alkoxy group donor.

Previously, we have developed the efficient protocols for ring-closing metathesis of esters **1** and **3a** [28].

Having an optimal reaction condition in hand for both studied transformations, next we tried to combine the enzymatic transesterification with subsequent ring-closing metathesis in a one-pot procedure. The main limitation of this step can be the chemoselectivity, since, besides the RCM reaction, the cross metathesis can occur concurrently. Our previous studies [28] indicate that RCM of vinyl 3-phenyl-4-pentenoate (**1a**) proceeds in a chemoselective manner, as the formation of dimers was not observed. However, in the studied case, the allyl alcohol and allyl 3-phenyl-4-pentenoate (**3a**) are also presented in the reaction mixture. We have also observed that RCM of the studied esters proceeds much faster than enzymatic transesterification; thus, the metathesis catalyst should be added to the reaction mixture after the completion of enzymatic transformation (Table 3). However, in this case, the enzyme and a solvent were not removed from the reaction mixture before the addition of Grubbs second-generation catalyst. The only change in the reaction condition was the addition of the metathesis catalyst and elevation of temperature to 80 °C. We have also tested the substrate scope for the studied one-pot, two-step synthesis (Table 3).

As a result of such a one-pot, two-step procedure, the mixture of enantiomerically pure (*S*)-4-phenyl-3,4-dihydro-2H-pyran-2-one (**4a**) and (*R*)-4-phenyl-3,4-dihydrooxepin-2(7H)-one (**5a**) were isolated with the yields of 47% and 43%, respectively (Table 3, entry 1). The application of 4-bromo- and 4-chlorophenyl derivatives (**1b** and **1c**) for a studied protocol leads to the formation of non-racemic mixtures of six- and seven-membered lactones (Table 3, entries 2–3). For any of lactones with *p*-chloro- substituent in an aromatic ring (**4c**, **5c**), enantiomeric excesses were not determined, and the values of optical rotations indicate that the obtained products were non-racemic (Table 1, entry 3). The results of the reaction of vinyl 3-propyl-4-penetnoate (**1d**) show that the developed protocol can be also applied for alkyl derivatives (Table 3, entry 4).

Encouraged by these results, in the next step we tested if the proposed EKR-RCM procedure would also be suitable for the simultaneous synthesis of lactones and lactams via a one-pot procedure. Thus, at first, we examined the enzymatic amidation of vinyl 3-phenyl-4-pentenoate (**1a**) with allylamine. However, for all the studied enzymes, the obtained amides were racemic. Thus, we tested N-allylbenzylamine (**6**). The polarity of the corresponding amide **7a** is significantly different than the polarity of ester **1a**; thus, these compounds are easily separable. Therefore, the reaction efficiency, as well as the enantiomeric purity of the obtained products, were measured directly after the completion of EKR. The reactions were carried in toluene at 70 °C for 24 h (Table 4). 

The reaction in the absence of enzyme did not procced (Table 4, entry 1). Both tested immobilized enzymes, Novozym and Lipozyme, catalyzed the studied transformation; however, the best result was obtained for Novozym—vinyl ester (*S*)-**1a** and amide (*R*)-**7a** were obtained in enantiopure forms (Table 4, entry 2). Eventually, we combined the developed enzymatic kinetic resolution protocol with ring-closing metathesis in a one-pot, two-step sequence to obtain enantiomerically pure six-membered lactone (*S*)-**4a** and lactam (*R*)-**8a** (Table 5, entry 1). It is worth mentioning that under the subsequent ring-closing metathesis reaction the racemization of esters and amides does not occur; both heterocycles were obtained as single enantiomers. The RCM reaction was also chemoselective as only cyclic products were obtained. We have also tested other vinyl esters (**1b–d**) in the studied one-pot simultaneous synthesis of lactones and lactams (Table 5, entries 2–4). 

The importance of our work was highlighted in the synthetic application of key intermediate **4a** for diversified synthesis of high-added-value compounds (Figure 2). As representative examples, we prepared 4-phenyltetrahydro-2H-pyran-2-one (**9**), a valuable building block [47] that can be further converted into other high-value-added compounds, such as azetidinone cholesterol absorption inhibitors [48,49] and others [50,51]. We also transformed lactone **4a** into methyl 5-oxo-3-phenylpentanoate (**10**), a precursor for serine proteases [52] and also a synthon for the preparation of Femoxetine, a serotonin reuptake inhibitor being studied in the treatment of narcolepsy, migraines, depressive states, and eating disorders [53,54], or as a potent renin [55] or BACE-1 [56] (used in Alzheimer disease treatment) inhibitors.

## 3. Materials and Methods

### 3.1. General

All the chemicals were obtained from commercial sources. The solvents were of analytical grade. NMR spectra were recorded in CDCl_3_ with TMS as an internal standard using Bruker 400 MHz spectrometers. The chemical shifts are reported in ppm (*δ* scale) and the coupling constants (*J*) are given in hertz (Hz). All the reactions were monitored by TLC on Merck silica gel Plates 60 F_254_. Column chromatography was performed on a Merck silica gel 60/230–400 mesh. 

### 3.2. General Procedure of the Enzymatic Kinetic Resolution Reaction (EKR) of 1a–d with 2

To the solution of vinyl 3-substituted-4-pentenoate **1a**–**d** (0.1 mmol) in toluene or other solvent (2 mL), the appropriate alkoxy group donor (**2**) (2 equiv.) and enzyme (10 mg) were added in a 5 mL screwed vial. The reaction mixture was stirred for 24–48 h at 40–70 °C. After cooling, the enzyme was filtered off and the solvent was evaporated. Then, to analyze the enantiomeric excesses, the products were transformed into corresponding lactones. The residue was dissolved in dry toluene (10 mL) under argon atmosphere and Grubbs second-generation catalyst was added (5 mol%). The reaction mixture was heated at 80 °C for 16 h. The reaction mixture was cooled to room temperature and filtered over a Celite pad. The excess of solvent was removed under reduced pressure. The crude product was purified by column chromatography on silica gel (hexanes/ethyl acetate).

### 3.3. General Procedure of the Enzymatic Kinetic Resolution Reaction (EKR) of 1a–d with 6

To the solution of vinyl 3-substituted-4-pentenoate **1a**–**d** (0.1 mmol) in toluene (2 mL), the appropriate allylamine (**6**) (2 equiv.) and enzyme (10 mg) were added in a 5 mL screwed vial. The reaction mixture was stirred for 24 h at 70 °C. After cooling, the crude product was purified by column chromatography (ethyl acetate/hexanes).

### 3.4. General Procedure of the Ring-Closing Metathesis Reaction (RCM)

To a solution of an olefin (0.4 mmol) in dry solvent (40 mL) under argon atmosphere was added Grubbs second-generation catalyst (5 mol%). The reaction mixture was heated at 80 °C for 16 h. The reaction mixture was cooled to room temperature and filtered over a Celite pad. The excess of solvent was removed under reduced pressure. The crude product was purified by column chromatography on silica gel (hexanes/ethyl acetate).

#### 3.4.1. 4-Phenyl-3,4-Dihydro-2H-Pyran-2-One (4a)

The product was isolated as a colorless oil from a silica column eluted by EtOAc/hexanes (1:9) in 96% yield (66 mg); ^1^H NMR (400 MHz, CDCl_3_): *δ* 7.31–7.20 (m, 3H), 7.17–7.12 (m, 2H), 6.61 (dd, *J* = 6.0, 1.8 Hz, 1H), 5.37 (dd, *J* = 6.0, 4.0 Hz, 1H), 3.79–3.69 (m, 1H), 2.91 (dd, *J* = 15.9, 6.5 Hz, 1H), 2.67 (dd, *J* = 16.0, 8.3 Hz, 1H).; ^13^C{H} NMR (100 MHz, CDCl_3_): δ 167.3, 141.6, 141.2, 129.1, 127.5, 126.8, 109.4, 37.3, 36.7; HRMS (ESI+, *m/z*): calcd. for C_11_H_11_O_2_ [M + H]^+^: 175.0759, found 175.0768. The ^1^H and ^13^C NMR data were in accordance with those reported in the literature [29].

#### 3.4.2. 4-(4-Bromophenyl)-3,4-Dihydro-2H-Pyran-2-One (4b)

The product was isolated as a colorless oil from a silica column eluted by EtOAc/hexanes (1:9) in 95% yield (96 mg); ^1^H NMR (200 MHz, CDCl_3_): *δ* 7.52–7.34 (m, 2H), 7.09–6.94 (m, 2H), 6.61 (dd, *J* = 6.0, 1.7 Hz, 1H), 5.33 (dd, *J* = 6.0, 4.1 Hz, 1H), 3.78–3.64 (m, 1H), 2.89 (dd, *J* = 15.9, 6.9 Hz, 1H), 2.61 (dd, *J* = 15.9, 8.0 Hz, 1H).; ^13^C{H} NMR (50 MHz, CDCl_3_): δ 169.2, 142.5, 142.0, 132.2, 125.6, 125.2, 121.4, 108.7, 37.1, 36.2; HRMS (EI+, *m/z*): calcd. for C_11_H_9_BrO_2_ [M]^+^: 251.9786, found 251.9741.

#### 3.4.3. 4-(4-Chlorophenyl)-3,4-Dihydro-2H-Pyran-2-One (4c)

The product was isolated as a colorless oil from a silica column eluted by EtOAc/hexanes (1:9) in 87% yield (72 mg); ^1^H NMR (200 MHz, CDCl_3_): δ 7.33–7.19 (m, 2H), 7.16–6.97 (m, 2H), 6.62 (dd, J = 6.0, 1.8 Hz, 1H), 5.34 (dd, J = 6.1, 4.1 Hz, 1H), 3.77–3.68 (m, 1H), 2.90 (dd, J = 16.0, 7.0 Hz, 1H), 2.62 (dd, J = 16.0, 8.0 Hz, 1H).; ^13^C{H} NMR (50 MHz, CDCl_3_): δ 171.2, 141.9, 141.5, 129.2, 128.2, 108.8, 37.2, 36.1; HRMS (EI+, *m*/*z*): calcd. for C_11_H_9_ClO_2_ [M]^+^: 208.0291, found 208.0287. 

#### 3.4.4. 4-Propyl-3,4-Dihydro-2H-Pyran-2-One (4d) 

The product was isolated as a colorless oil from a silica column eluted by EtOAc/hexanes (1:9) in 83% yield (46 mg); ^1^H NMR (400 MHz, CDCl_3_): δ 6.40 (dd, J = 6.0, 1.6 Hz, 1H), 5.18 (dd, J = 5.9, 4.0 Hz, 1H), 2.69–2.59 (m, 1H), 2.51–2.42 (m, 1H), 2.34 (dd, J = 15.5, 8.0 Hz, 1H), 1.38–1.27 (m, 4H), 0.88–0.81 (m, 3H); ^13^C{H} NMR (100 MHz, CDCl_3_) δ 168.5, 140.6, 110.4, 36.8, 34.9, 30.2, 19.5, 13.9; HRMS (ESI+, *m*/*z*): calcd. for C_8_H_12_O_2_ [M + H]^+^: 140.0837, found 140.0849. The ^1^H and ^13^C NMR data were in accordance with those reported in the literature [29].

#### 3.4.5. 4-Phenyl-3,4-Dihydrooxepin-2(7H)-One (5a)

The product was isolated from a silica column eluted by EtOAc/hexanes (2:8) in 75% yield (56 mg) as white crystals; ^1^H NMR (400 MHz, CDCl_3_): *δ* 7.40–7.25 (m, 5H), 6.08–5.99 (m, 1H), 5.95–5.85 (m, 1H), 4.99–4.93 (m, 1H), 4.71–4.65 (m, 1H), 3.95–3.86 (m, 1H), 3.18 (dd, *J* = 13.0, 10.2 Hz, 1H), 3.04 (ddd, *J* = 13.1, 4.2, 1.0 Hz, 1H); ^13^C NMR (100 MHz, CDCl_3_): δ 172.8, 141.8, 135.5, 128.9, 127.5, 127.4, 124.5, 63.9, 42.3, 39.7. The ^1^H and ^13^C NMR data were in accordance with those reported in the literature [28].

#### 3.4.6. 4-(4-Bromophenyl)-3,4-Dihydrooxepin-2(7H)-One (5b)

The product was isolated from a silica column eluted by EtOAc/hexanes (2:8) in 62% yield (32 mg) as white crystals; ^1^H NMR (400 MHz, CDCl_3_): *δ* 7.40 (d, *J* = 8.4 Hz, 2H), 7.03 (d, *J* = 8.4 Hz, 2H), 5.98–5.92 (m, 1H), 5.82–5.76 (m, 1H), 4.84–4.71 (m, 1H), 4.68–4.56 (m, 1H), 3.83–3.75 (m, 1H), 3.07–2.88 (m, 2H); ^13^C NMR (100 MHz, CDCl_3_): δ 168.1, 146.1, 134.7, 132.1, 129.2, 125.1, 63.8, 41.7, 39.5. HRMS (EI+, *m/z*): calcd. for C_12_H_11_BrO_2_ [M]^+^: 267.1185, found 267.1187.

#### 3.4.7. 4-(4-Chlorophenyl)-3,4-Dihydrooxepin-2(7H)-One (5c)

The product was isolated from a silica column eluted by EtOAc/hexanes (2:8) in 58% yield (25 mg) as white crystals; ^1^H NMR (400 MHz, CDCl_3_): *δ* 7.25–7.20 (m, 2H), 7.11–7.08 (m, 2H), 5.98–5.94 (m, 1H), 5.79–5.75 (m, 1H), 4.84–4.79 (m, 1H), 4.64–4.56 (m, 1H), 3.75–3.71 (m, 1H), 3.01–2.97 (m, 2H); ^13^C NMR (100 MHz, CDCl_3_): δ 171.1, 142.3, 134.6, 131.6, 129.8, 128.1, 63.5, 43.5, 39.9. HRMS (EI+, *m/z*): calcd. for C_12_H_11_ClO_2_ [M]^+^: 222.6679, found 222.6680.

#### 3.4.8. 4-Propyl-3,4-Dihydrooxepin-2(7H)-One (5d)

The product was isolated from a silica column eluted by EtOAc/hexanes (2:8) in 43% yield (13 mg) as semi-solid; ^1^H NMR (400 MHz, CDCl_3_): 5.75–5.65 (m, 2H), 4.64–4.49 (m, 2H), 2.84–2.74 (m, 1H), 2.71–2.69 (m, 1H), 2.52–2.48 (m, 1H), 1.40–1.34 (m, 4H), 0.88–0.84 (m, 3H); ^13^C NMR (100 MHz, CDCl_3_): δ 172.3, 141.1, 132.4, 65.0, 40.1, 36.7, 20.1, 14.0. HRMS (EI+, *m*/*z*): calcd. for C_9_H_14_O_2_ [M]^+^: 154.2066, found 154.2068. 

#### 3.4.9. 1-Benzyl-4-Phenyl-1,3,4,7-Tetrahydro-Azepin-2-One (8a) 

The product was isolated from silica column eluted by EtOAc/hexanes (25:75) with 91% yield (101 mg) as white crystals; ^1^H NMR (400 MHz, CDCl_3_): *δ* 7.30–7.11 (m, 10H), 5.77–5.62 (m, 2H), 4.59 (s, 2H), 4.16 (ddt, *J* = 17.7, 4.1, 2.4 Hz, 1H), 3.77 (dt, *J* = 11.6, 3.0 Hz, 1H), 3.51 (ddd, *J* = 17.6, 6.6, 1.6 Hz, 1H), 3.08 (dd, *J* = 13.0, 11.4 Hz, 1H), 2.79–2.70 (m, 1H); ^13^C NMR (100 MHz, CDCl_3_): δ 172.8, 143.7, 137.4, 134.8, 128.7, 128.5, 128.0, 127.5, 127.4, 126.9, 124.9, 51.2, 45.2, 42.2, 42.1; HRMS (ESI+, *m/z*): calcd. for C_19_H_20_NO [M + H]^+^: 278.1539, found 278.1536. The ^1^H and ^13^C NMR data were in accordance with those reported in the literature [57].

#### 3.4.10. 1-Benzyl-4-(4-Bromophenyl)-1,3,4,7-Tetrahydro-Azepin-2-One (8b)

The product was isolated from silica column eluted by EtOAc/hexanes (25:75) with 80% yield (42 mg) as white crystals; ^1^H NMR (400 MHz, CDCl_3_): *δ* 7.40–7.31 (m, 2H), 7.26–7.17 (m, 5H), 7.09–7.03 (m, 2H), 5.98–5. 96 (m, 1H), 5.48–5.42 (m, 1H), 4.60 (s, 2H), 3.46–3.41 (m, 1H), 2.78 (d, *J* = 6.7 Hz, 2H), 2.42–2. 34 (m, 1H), 2.16–2.08 (m, 1H); ^13^C NMR (100 MHz, CDCl_3_): δ 172.2, 144.1, 131.7, 130.8, 128.6, 128.3, 127.9, 127.4, 50.3, 45.9, 42.4, 33.6; HRMS (EI+, *m*/*z*): calcd. for C_19_H_18_BrNO [M]^+^: 356.2566, found 356.2569. 

#### 3.4.11. 1-Benzyl-4-(4-Chloroophenyl)-1,3,4,7-Tetrahydro-Azepin-2-One (8c) 

The product was isolated from silica column eluted by EtOAc/hexanes (25:75) with 75% yield (31 mg) as white crystals; ^1^H NMR (400 MHz, CDCl_3_): *δ* 7.32–7.25 (m, 7H), 7.19–7.17 (m, 2H), 6.07–6.04 (m, 1 H), 5.55–5.49 (m, 1H), 4.67 (s, 2H), 3.56–3.49 (m, 1H), 2.86 (d, *J* = 6.8 Hz, 2H), 2.49–2.42 (m, 1H), 2.26–2.15 (m, 1H).; ^13^C NMR (50 MHz, CDCl_3_): δ 172.3, 143.6, 137.3, 130.7, 128.7, 128.6, 128.0, 127.9, 127.4, 50.3, 45.9, 42.5, 33.6. HRMS (EI+, *m/z*): calcd. for C_19_H_18_ClNO [M]^+^: 311.8060, found 311.8058. 

#### 3.4.12. 1-Benzyl-4-Propyl-1,3,4,7-Tetrahydro-Azepin-2-One (8d)

The product was isolated from silica column eluted by EtOAc/hexanes (25:75) with 62% yield (15 mg) as white crystals; ^1^H NMR (400 MHz, CDCl_3_): *δ* 7.30–7.14 (m, 5H), 5.61–5.57 (m, 1H), 5.56–5.44 (m, 1H), 4.55 (s, 2H), 3.88–3.74 (m, 1H), 3.61–3.54 (m, 1H), 2.72–2.63 (m, 2H), 2.51–2.38 (m, 1H), 1.41–1.32 (m, 4 H), 0.87–0.81 (m, 3H); ^13^C NMR (100 MHz, CDCl_3_): δ 173.7, 137.5, 136.3, 128.4, 128.0, 127.8, 127.2, 123.5, 51.0, 45.4, 38.8, 38.1, 35.2, 19.7, 14.0.; HRMS (EI+, *m/z*): calcd. for C_16_H_21_NO [M]^+^: 243.3447, found 243.3450.

### 3.5. General Procedure of the One-Pot EKR-RCM

The mixture of vinyl 3-substituted-4-pentenoate **1a**–**d** (0.2 mmol), donor (**2** or **6**) (2 equiv.) and Novozym (10 mg) was stirred in dry toluene (10 mL) at 70 °C. After 24 h (for **6**) or 48 h (for **2**) Grubbs second-generation catalyst (5 mol%) was added. The reaction mixture was heated at 80 °C for 16 h under argon atmosphere. The reaction mixture was cooled to room temperature and filtered over a Celite pad. The excess of solvent was removed under reduced pressure. The crude product was purified by column chromatography on silica gel (hexanes/ethyl acetate).

Further experimental procedures and data are given in the Appendix A.

## 4. Conclusions

In conclusion, a new concept for a stereoselective diversity-oriented synthesis based on the one-pot combination of kinetic resolution with subsequent transformation was introduced. The proposed approach overcomes the limitations of the application of kinetic resolution in DOS and extends the possible library of diverse compounds twice in shorter time. Two types of structurally and enantiodivergent products were obtained, which can serve as platform molecules in subsequent DOS reactions. To the best of our knowledge, this is the first example of time- and cost-effective application of kinetic resolution reactions in DOS. The proposed concept was proved in the enzymatic kinetic resolution/ring-closing metathesis cascade. As products, two enantiomerically pure, structurally diverse lactones and lactams with distinct configurations were obtained. It is worth mentioning that the subsequent ring-closing metathesis reaction is chemoselective; as only cyclic products were obtained, the formation of cross metathesis products was not observed, and under the reaction conditions the racemization of substrates and products does not occur. To the best of our knowledge, the presented strategy is the first example of a metathesis reaction where two various substrates undergo RCM simultaneously in a one-pot, leading to the formation of two distinct heterocycles exclusively (without the formation of cross-metathesis products). As the whole starting material is converted into valuable heterocycles, the presented methodology fulfils the principles of green chemistry metrics, such an atom economy and *E*-factor. The developed cascade can be employed as a key step toward the synthesis of an enantiodivergent library of structurally diverse heterocycles such as active pharmaceutical ingredients (API).

## Data Availability

The data presented in this study are available on request from the corresponding author.

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
