# Peer review of "Simultaneous Enantiodivergent Synthesis of Diverse Lactones and Lactams via Sequential One-Pot Enzymatic Kinetic Resolution–Ring-Closing Metathesis Reactions"

_molecules, 2022, doi:10.3390/molecules27227696_

Round 1

Reviewer 1 Report

Research paper by Anna Brodzka et al for enantiodivergent synthesis of lactones and lactams using EKR-RCM cascade has prominent scope to develop novel enantioselective molecules with applications in drug discovery. Overall, this article fit the aim & scope of the journal "Molecules". This article reports an improvement for the previous method by same authors using kinetic resolution to develop enantiomers. 

Comments:

Supporting information:

13C NMR of compounds 7b, 7c, 7d, 8b, 8c, 8d are too noisy and hard to differentiate 13C NMR peaks and noise peaks. Reviewer suggest authors to include decent 13C NMR spectra for the above-mentioned compounds.

1H NMR for 1b, 1c, 3a, 3b, 3c, 4b, 4c, 7c, 8b, 8c, 8d, 9, 10 have extra NMR peaks apart from the product peaks, how much pure the above compounds. Reviewer suggest authors to include clean 1H NMR spectra for the above compounds after the purification. Peak at TMS (0 ppm) and peak around 1.5 ppm (for water) is fine, but other peaks should be avoided or need to provide explanation what those peaks are?

Line 38: It should be and not ad.

Author Response

Rewiever 1:

  1. Supporting information:

13C NMR of compounds 7b, 7c, 7d, 8b, 8c, 8d are too noisy and hard to differentiate 13C NMR peaks and noise peaks. Reviewer suggest authors to include decent 13C NMR spectra for the above-mentioned compounds.

Answer: We are grateful for this opinion. The spectra of the mentioned compounds were now recorded on 400 MHz spectrometer (previously they were recorded on 200MHz) and included in SI.

  1. 1H NMR for 1b, 1c, 3a, 3b, 3c, 4b, 4c, 7c, 8b, 8c, 8d, 9, 10 have extra NMR peaks apart from the product peaks, how much pure the above compounds. Reviewer suggests authors to include clean 1H NMR spectra for the above compounds after the purification. Peak at TMS (0 ppm) and peak around 1.5 ppm (for water) is fine, but other peaks should be avoided or need to provide explanation what those peaks are?

Answer: We are grateful for this opinion. The mentioned compounds were additionally purified and the new spectra (400 MHz) were recorded. The other peaks are impurities from metathesis catalyst and/or residual solvents.

  1. Line 38: It should be and not ad.

Answer: The found errors were corrected.

Reviewer 2 Report

In this work an enzymatic kinetic resolution of vinyl esters during transesterification and amidation reaction to render the corresponding O-allylic esters and N-allylic amides, respectively, (maximum 50% yield) in addition to unreacted vinyl ester, is described. This transformation is then demonstrated compatible with a one-pot Ru-catalyzed ring closing metathesis to lactones/lactams. Thus, the two-reaction sequence (one enzymatic and one metal-catalyzed) delivers a mixture of enantioenriched 6- and 7-membered unsaturated lactones, or, alternatively, a mixture of 6-membered lactone and 7-membered lactam. Authors present these results as an example of diversity oriented synthesis although this aspect is not fully convincing based on the few experiments reported. The suitability of both catalytic reaction conditions to merge one-pot is interesting and application of this strategy in different catalytic reactivity settings may be foreseen. However, the actual scope demonstration is a little bit limited, with only four entries for the transesterification approach (out of which only the Ph-substituted entry gives really high enantioselectivities) and four additional for the lactamization approach (again, with only one truly highly enantioselective example).

My feeling is that the work deserves publication once some additional entries with high enantioselectivity are collected.

Before publication, erasing the word “Cascade” from the title is highly recommended, as this term implies some requirements that the current method does not meet. “Sequential One-Pot” may be a better alternative.

Also, some citations in the references section need revision. For instance, article title in Ref 19 is missing and journal name (English version) and citation data in Ref 3 need adjustment.

Author Response

Rewiever 2:

  1. Before publication, erasing the word “Cascade” from the title is highly recommended, as this term implies some requirements that the current method does not meet. “Sequential One-Pot” may be a better alternative.

Answer: We are grateful for this opinion. The title was changed on: “Simultaneous Enantiodivergent Synthesis of Diverse Lactones and Lactams via Sequential One-Pot Enzymatic Kinetic Resolution – Ring-Closing Metathesis Reactions.”

  1. Also, some citations in the references section need revision. For instance, article title in Ref 19 is missing and journal name (English version) and citation data in Ref 3 need adjustment.

Answer:  The found errors were corrected.

Round 2

Reviewer 1 Report

Quality of the paper was improved after the revision. The correct spelling error at Line 38 (should be and not ad) was not included in the revised version. Authors need to correct it.

Reviewer 2 Report

I believe the manuscript has been sufficiently improved to warrant publication in Molecules